# A CFO-Assisted Algorithm for Wireless Time-Difference-of-Arrival Localization Networks: Analytical Study and Experimental Results

**DOI:** 10.3390/s24030737

**Published:** 2024-01-23

**Authors:** Cédric Hannotier, François Horlin, François Quitin

**Affiliations:** Brussels School of Engineering, Université libre de Bruxelles, Avenue Franklin Roosevelt 50, 1000 Brussels, Belgium; francois.horlin@ulb.be (F.H.); francois.quitin@ulb.be (F.Q.)

**Keywords:** wireless systems, localization, Time Difference of Arrival (TDoA) estimation, Over-The-Air (OTA) synchronization

## Abstract

Localization of wireless transmitters is traditionally done using Radio Frequency (RF) sensors that measure the propagation delays between the transmitter and a set of anchor receivers. One of the major challenges of wireless localization systems is the need for anchor nodes to be time-synchronized to achieve accurate localization of a target node. Using a reference transmitter is an efficient way to synchronize the anchor nodes Over-The-Air (OTA), but such algorithms require multiple periodic messages to achieve tight synchronization. In this paper, we propose a new synchronization method that only requires a single message from a reference transmitter. The main idea is to use the Carrier Frequency Offset (CFO) from the reference node, alongside the Time of Arrival (ToA) of the reference node messages, to achieve tight synchronization. The ToA allows the anchor nodes to compensate for their absolute time offset, and the CFO allows the anchor nodes to compensate for their local oscillator drift. Additionally, using the CFO of the messages sent by the reference nodes and the target nodes also allow us to estimate the speed of the targets. The error of the proposed algorithm is derived analytically and is validated through controlled laboratory experiments. Finally, the algorithm is validated by realistic outdoor vehicular measurements with a software-defined radio testbed.

## 1. Introduction

Accurate localization of Radio Frequency (RF) transmitters is an important feature of future wireless networks, with applications in smart cities, industry 4.0 and vehicular technology [1]. Localization through wireless networks is especially important when Global Navigation Satellite Systems (GNSSs) are unavailable or unreliable, for instance, in urban or suburban environments due to severe multipath conditions or blockage of satellite signals [1,2,3,4,5].

Different features of RF signals can be used for localization, among which are Received Signal Strenght (RSS), Angle of Arrival (AoA), Time of Arrival (ToA) or Time Difference of Arrival (TDoA) [6]. In a TDoA-based localization system, several anchor nodes (with known locations) estimate the difference in ToA of a signal transmitted by a target node. The TDoA between a pair of anchor nodes defines a hyperbola of possible target node locations, and the intersection of multiple such hyperbola indicates the target location. TDoA-based localization has multiple advantages: it does not require costly multi-antenna arrays, the target node does not need to be synchronized with the anchor nodes and the target node need not be cooperative with the anchor nodes [3]. However, one important prerequisite of TDoA-based localization is that the anchor nodes need to be accurately time-synchronized to achieve precise localization. Each anchor node has its own Local Oscillator (LO), and clock parameters (commonly referred to as clock offset and drift) differ randomly at each node. Without synchronization, each receiver would utilize a different time reference when measuring the ToA of the target message, resulting in a substantial amount of error when estimating the TDoA. As a result, the synchronization of receivers is essential in TDoA estimation [1].

Synchronization methods for anchor nodes can generally be divided into two categories: wired and wireless synchronization. In a wired synchronization network, a reference signal is transmitted over cables that can be used for synchronizing the anchor nodes. Advanced LO tracking algorithms have been implemented that can synchronize oscillators down to the nanosecond over coaxial cables or over fiber optics (such as the White Rabbit project protocol [7]). However, wired synchronization requires expensive deployments, which may not be feasible in practical situations. Wireless synchronization can be subdivided into Global Navigation Satellite Systems (GNSS) and Over-The-Air (OTA) synchronization methods. GNSS synchronization requires equipping each anchor node with a GNSS receiver module, which can use the Universal Time Coordinate (UTC) as a reference time and yield accuracies from a few nanoseconds to several tens of nanoseconds [8]. However, the synchronization accuracy will severely be degraded in indoor environments, urban environments or tunnels due to limited GNSS visibility. In such environments, the accuracy goes to hundreds of nanoseconds or more, making GNSS synchronization unbearable. OTA synchronization methods are usually based on a broadcaster node, which is used to distribute reference signals [9,10]. The drawback of OTA synchronization is that some bandwidth capacity is used to distribute the synchronization information, and the resulting accuracy is highly dependent on the rate of the synchronization messages from the broadcaster node [9].

In our preliminary work [11], we introduced a new OTA one-way synchronization algorithm for a TDoA system with a broadcaster. It is based on the realization that the time skew and the Carrier Frequency Offset (CFO) between a transmitter and a receiver are related. By using CFO estimations, we are able to achieve better time synchronization over time from a single message from the broadcaster, allowing a lower synchronization message rate. In this paper, we extend our previous work to include an error analysis of the proposed algorithm, extensive outdoor measurements in a vehicular scenario and speed estimation.

**Contributions:** The contributions are listed as follows:We propose a CFO-assisted TDoA localization algorithm, which is able to synchronize the anchor nodes with only a single message from a broadcaster node;On top of localization information, our proposed method is also able to provide information about the movement of the target, even with a single message from the target node;The error of the proposed algorithms is investigated analytically and a closed-form solution is proposed;A set of controlled lab experiments is realized to validate our previous findings;An extensive outdoor vehicular measurement campaign is conducted to validate and quantify the performances of the proposed algorithms.

The significance of the proposed algorithms lies in the fact that we are able to perform TDoA-based localization with *only a single message with no time metadata from the broadcaster*, as opposed to other synchronization algorithms that require multiple messages from a broadcaster node to achieve accurate synchronization.

The paper is organized as follows: Section 2 introduces the algorithms proposed in this paper, while Section 3 presents and analyzes experimental results. In particular, Section 2 is structured as follows: Section 2.1 introduces the TDoA system, its need for synchronization and the clock model. Section 2.2 and Section 2.3 present some conventional synchronization methods for TDoA estimation. In Section 2.4, the proposed CFO-assisted synchronization algorithm is presented and its accuracy is analytically evaluated. The estimation is then extended to include a velocity estimator.

The following notations are used in this paper: ^*T*^, ‖ ‖ are defined as the matrix transpose and the Euclidean norm, respectively. Moreover, vectors and matrices are bold.

## 2. Materials and Methods

### 2.1. System Model

#### 2.1.1. TDoA Architecture

The TDoA system consists of N-receivers with known positions 
$p_{i}={\left[x_{i},y_{i}\right]}^{T}$
 (
i=0,…,N−1
), and a target at an unknown position 
$p_{T}={\left[x_{T},y_{T}\right]}^{T}$
, as depicted in Figure 1 (the broadcaster node will be introduced later).The target transmits RF packets at a known carrier frequency. The receivers record their ToAs: 
(1)
$$t_{i}=\underset{\tau_{i}}{\underset{︸}{∥p_{i}-p_{T}∥/c}}+t_{T}$$

where 
$\tau_{i}$
 is the propagation delay between the target and the receiver *i*, *c* the speed of transmission through the medium and 
$t_{T}$
 the unknown transmission time. From these ToAs, a TDoA system of equations to recover the target location is constructed, where the TDoA 
$\tau_{ij}$
 between receivers *i* and *j* is defined as: 
(2)
$$\begin{array}{cc} t_{j}-t_{i} & =(∥p_{j}-p_{T}∥-∥p_{i}-p_{T}∥)/c\\ \; & =\tau_{j}-\tau_{i}\equiv \tau_{ij}. \end{array}$$


In this paper, only additive measurement noise on the estimation of 
$t_{i}$
 is considered, neglecting effects due to the environment, such as multipath components or loss of line-of-sight. A TDoA localization method robust to such impairments might be required, as they will degrade the TDoA estimation. For clarity purposes, the additive noise is not shown hereafter.

#### 2.1.2. LO Model

To compute timing information, as in (Equation 1), devices use LOs. These LOs are non ideal, hence locally distorting the notion of time. A common model for these imperfections is [9,12,13]:
(3)
$$T_{i}\left(t\right)=\alpha_{i}+\beta_{i}t,$$

where 
$T_{i}\left(t\right)$
 is the time recorded by the LO of node *i* at the absolute time reference *t*, and depends on an initial shift parameter 
$\alpha_{i}$
 and a drift parameter 
$\beta_{i}$
. These clock parameters are usually slowly varying in time. However, they are varying much slower than the regular period (<1 s) between two synchronization updates. These clock parameters are hence assumed constant.

On many devices, the ADC and the down-converter are driven by the same LO, as shown in Figure 2. Since the LO is not perfect, it will lock within an offset of the nominal frequency. In such cases, the frequencies of the clocks driving the ADC 
$f_{s_{i}}$
 and the down-converter 
$f_{c_{i}}$
 can be written as follows:
(4)
$$\begin{array}{cc} f_{c_{i}} & =f_{c}\left(1+\epsilon_{i}\right),\\ f_{s_{i}} & =f_{s}\left(1+\epsilon_{i}\right), \end{array}$$

and the drift parameter 
$\beta_{i}$
 can be rewritten as:
(5)
$$\beta_{i}=1+\epsilon_{i}.$$


Hence, the ToA (Equation 1) measured by anchor *i* of a message sent by the target at absolute time 
$t_{T}$
 becomes:
(6)
$$\begin{array}{cc} T_{i}\left(t_{T}+\tau_{i}\right) & =\alpha_{i}+\left(1+\epsilon_{i}\right)\left(t_{T}+\tau_{i}\right)\\ \; & \approx \alpha_{i}+\left(1+\epsilon_{i}\right)t_{T}+\tau_{i}. \end{array}$$


In the final equation, the second-order term 
$\epsilon_{i}\tau_{i}$
 is an order of magnitude smaller than the others and can therefore be neglected (
$\epsilon_{i}$
 is typically in the order of parts-per-million, i.e., 1 × 10^−6^, and 
$\tau_{i}$
 is typically in the order of tens to hundreds of nanoseconds).

### 2.2. TDoA without Synchronization

Using this model, without any prior synchronization, the estimated TDoA between two receivers is given by:
(7)
$$\tau_{ij}^{NS}=T_{j}\left(t_{T}+\tau_{j}\right)-T_{i}\left(t_{T}+\tau_{i}\right)=\left(\alpha_{j}-\alpha_{i}\right)+\left(\epsilon_{j}-\epsilon_{i}\right)t_{T}+\tau_{ij}.$$


In this case, the TDoA 
$\tau_{ij}$
 is tainted by a constant offset 
$\left(\alpha_{j}-\alpha_{i}\right)$
 and drifts over time with a factor 
$\left(\epsilon_{j}-\epsilon_{i}\right)$
.

### 2.3. Broadcaster-Assisted TDoA

To compensate for the constant offset and time drift, one can use other node broadcasting messages (denoted the broadcaster, depicted in Figure 1), and use these messages as an absolute time reference. Without loss of generality, we will assume that the propagation delay between the broadcaster and anchor *i*

$\tau_{B_{i}}=0$
. In practice, this means that the propagation delay between the broadcaster and the anchors needs to be estimated and compensated for. The broadcaster-assisted ToA can then be defined as:
(8)
$$\begin{array}{cc} t_{i}^{BS}\equiv T_{i}\left(t_{T}+\tau_{i}\right)-T_{i}\left(t_{B}\right) & =\left(1+\epsilon_{i}\right)\left(t_{T}-t_{B}\right)+\tau_{i}\\ \; & =\left(1+\epsilon_{i}\right)\Delta t+\tau_{i}, \end{array}$$

where 
$t_{B}$
 is the ToA of the last message received from the broadcaster and 
$\Delta t=t_{T}-t_{B}$
. The TDoA is then given by:
(9)
$$\tau_{ij}^{BS}\equiv t_{j}^{BS}-t_{i}^{BS}=\left(\epsilon_{j}-\epsilon_{i}\right)\Delta t+\tau_{ij}.$$


It can be seen that the constant offset is removed, and the drift is reset each time a new broadcasting message arrives. Hereafter, this method will be referred to as the broadcaster-assisted synchronization method. In the remainder of this paper, and without loss of generalization, we will define the absolute time to be the time that is estimated locally at the broadcast node, such that 
$\alpha_{B}=0$
, 
$\epsilon_{B}=0$
 and 
$\beta_{B}=1$
.

### 2.4. CFO-Assisted TDoA and Velocity Estimation

#### 2.4.1. CFO-Assisted TDoA Estimate

In Equation (Equation 9), the estimated TDoA is tainted by an error 
$(\epsilon_{j}-\epsilon_{i})\Delta t$
, which drifts when the time between two broadcast messages increases. In this paper, we propose to enhance the broadcaster-assisted synchronization method by using the CFO of the target message to compensate for the existing drift between two broadcast messages. Knowing the model (Equation 4), the CFO between the target and the receiver is:
(10)
$${\Delta f_{c}}_{i}=\left(\epsilon_{T}-\epsilon_{i}\right)f_{c}.$$


The proposed enhanced ToA estimate 
$t_{i}^{CS}$
 is the following:
(11)
$$\begin{array}{cc} t_{i}^{CS}\equiv t_{i}^{BS}\left(1+\frac{{\Delta f_{c}}_{i}}{f_{c}}\right)\; & =\left(\left(1+\epsilon_{i}\right)\Delta t+\tau_{i}\right)\left(1+\epsilon_{T}-\epsilon_{i}\right)\\ \; & =\left(1+\epsilon_{T}+\epsilon_{i}\epsilon_{T}-\epsilon_{i}^{2}\right)\Delta t+\left(1+\epsilon_{T}-\epsilon_{i}\right)\tau_{i}\\ \; & \approx \left(1+\epsilon_{T}\right)\Delta t+\tau_{i}. \end{array}$$


In the last equation, the second-order terms are an order of magnitude smaller than the others and can therefore be neglected (values for 
ϵ
 are typically in the order of parts-per-million, i.e., 1 × 10^−6^, and values for propagation delays 
$\tau_{i}$
 are typically in the tens or hundreds of nanoseconds). When using these “compensated” ToA, the corresponding TDoA becomes:
(12)
$$\tau_{ij}^{CS}\equiv t_{j}^{CS}-t_{i}^{CS}=\tau_{ij}.$$


It can be seen that both the offset and the drift have been compensated from the estimated TDoA.

Note that it is also possible to use the CFO of the broadcaster message to compensate for the ToA and TDoA values. The CFO is:
(13)
$${\Delta f_{c_{B}}}_{i}=-\epsilon_{i}f_{c},$$

while the compensated ToA 
$t_{i}^{C_{B}S}$
 is:
(14)
$$\begin{array}{cc} t_{i}^{C_{B}S}\equiv t_{i}^{BS}\left(1+\frac{{\Delta f_{c_{B}}}_{i}}{f_{c}}\right)\; & =\left(\left(1+\epsilon_{i}\right)\Delta t+\tau_{i}\right)\left(1-\epsilon_{i}\right)\\ \; & \approx \Delta t+\tau_{i}. \end{array}$$


The corresponding TDoA is then:
(15)
$$\tau_{ij}^{C_{B}S}\equiv t_{j}^{C_{B}S}-t_{i}^{C_{B}S}=\tau_{ij}.$$


In the remainder of the paper, unless stated otherwise, the CFO from the target will be used for estimating the CFO-assisted TDoA.

An example of TDoA error with the different synchronization methods is shown in Figure 3. Using the broadcaster-assisted method, the TDoA error increases at the same rate as without any synchronization. The accumulated error of the broadcaster-assisted method is compensated for each time a new broadcast packet is received (every second). The slope of that drift depends on the difference of the normalized frequency shifts 
$\epsilon_{Rx}$
 between receivers. Using the CFO-assisted method, the error is reduced between broadcast messages. In this case, the drift depends on the accuracy of the CFO estimation. The synchronization using the CFO from the target seems more noisy than the one using the CFO from the broadcaster because of the higher number of realizations of the former. Over two broadcaster messages, several CFOs from the target are estimated, but only one from the broadcaster.

#### 2.4.2. Variance of TDoA Error

The variance of the CFO-assisted estimator 
$\sigma_{\tau_{ij}^{CS}}^{2}$
 based on (Equation 11) and (Equation 12) is derived in Appendix A. It is given by: 
(16)
$$\sigma_{\tau_{ij}^{CS}}^{2}=2\left(\frac{\sigma_{{\Delta f_{c}}_{i}}^{2}}{f_{c}^{2}}{\bar{t_{i}^{BS}}}^{2}+\frac{\sigma_{t_{i}^{BS}}^{2}\sigma_{{\Delta f_{c}}_{i}}^{2}}{f_{c}^{2}}+\sigma_{t_{i}^{BS}}^{2}{\left(1+\frac{\bar{{\Delta f_{c}}_{i}}}{f_{c}}\right)}^{2}\right)-2\frac{{\bar{t_{i}^{BS}}}^{2}}{f_{c}^{2}}\mathrm{Cov}\left[{\Delta f_{c}}_{i},{\Delta f_{c}}_{j}\right].$$

where 
$\sigma_{{\Delta f_{c}}_{i}}^{2}$
 is the variance of the CFO estimation at anchor *i*, 
$\sigma_{t_{i}^{BS}}^{2}$
 is the variance of the ToA estimation at anchor *i*, 
$\bar{{\Delta f_{c}}_{i}}$
 is the estimated CFO and 
$\bar{t_{i}^{BS}}$
 is the estimated, broadcaster-compensated, ToA. It also includes the covariance between the CFO estimated at both receivers. Since that covariance can be challenging to estimate, and since it is always positive (see Appendix B), an upper bound estimate is used instead:
(17)
$$\sigma_{\tau_{ij}^{CS}}^{2}\leq 2\left(\frac{\sigma_{{\Delta f_{c}}_{i}}^{2}}{f_{c}^{2}}{\bar{t_{i}^{BS}}}^{2}+\frac{\sigma_{t_{i}^{BS}}^{2}\sigma_{{\Delta f_{c}}_{i}}^{2}}{f_{c}^{2}}+\sigma_{t_{i}^{BS}}^{2}{\left(1+\frac{\bar{{\Delta f_{c}}_{i}}}{f_{c}}\right)}^{2}\right).$$


The upper bound (Equation 17) is composed of three terms. The second and third terms are constant and depend mainly on the estimators used in the localization system. The first term increases with 
$\bar{t_{i}^{BS}}$
, i.e., the delay between the target and broadcast messages. This implies that when the time between a target and broadcast message becomes larger, a larger error will be incurred, even with the CFO-assisted TDoA estimation.

Simulation results and experimental results in a controlled experiment show that this upper bound is especially tight at lower Signal to Noise Ratios (SNRs), as will be shown in Section 3.

#### 2.4.3. Localization

The main goal of the TDoA system is the extraction of the target location. The estimation of the target location is based on the resolution of (Equation 2), which consists of a set of hyperbolas intersecting at the target location. Several algorithms have been proposed to solve these types of problems, such as [14,15,16,17]. In this paper, the localization performance is evaluated on outdoor measurements (see Section 2.5.3 for details). Since these measurements are tainted by outliers, the algorithm described in [18] is used. This algorithm was developed based on these measurements.

#### 2.4.4. Velocity Estimation

In the previous equations, velocity was not considered. If the target is moving, it induces a Doppler frequency, adding up to the CFO estimate. In such cases, the CFO estimate contains the contribution due to imperfect LOs 
${\Delta f_{c}}_{i}$
, and the contribution due to the movement of the target with regard to the receiver 
${F_{D}}_{i}$
: 
(18)
$$\begin{array}{c} CFO_{i}={\Delta f_{c}}_{i}-{F_{D}}_{i},\\ {F_{D}}_{i}=\frac{{\left(p_{T}-p_{i}\right)}^{T}v_{T}}{\lambda ∥p_{T}-p_{i}∥}, \end{array}$$

where 
$v_{T}={\left[{v_{T}}_{x},{v_{T}}_{y}\right]}^{T}$
 is the velocity of the target and 
λ
 is the wavelength of the RF wave.

Knowing the location of the target 
$p_{T}$
 is not sufficient to estimate its velocity from the system of equations generated from (Equation 18). Indeed, for a TDoA system with *N* anchors, (Equation 18) spawns *N* equations, but there are 
N+2
 unknowns (
${v_{T}}_{x}$
, 
${v_{T}}_{y}$
 and 
${\Delta f_{c}}_{i},\;i=1,\ldots ,N$
). Nonetheless, the CFO between the receiver and the broadcaster can be used to remove the dependency on the unknowns 
${\Delta f_{c}}_{i}$
. Since there is no Doppler frequency between the receiver and the broadcaster, the measured CFO is

(19)
$${CFO_{B}}_{i}={{\Delta f_{c}}_{B}}_{i}$$

and knowing that

(20)
$${\Delta f_{c}}_{i}-{{\Delta f_{c}}_{B}}_{i}=\epsilon_{T}f_{c}={\Delta f_{c}}_{T},$$

then another system of equations can be constructed: 
(21)
$${\Delta \mathrm{CFO}}_{i}\equiv CFO_{i}-{CFO_{B}}_{i}={\Delta f_{c}}_{T}-{F_{D}}_{i},$$

which reduces the unknowns to 
${v_{T}}_{x},{v_{T}}_{y}$
 and 
${\Delta f_{c}}_{T}$
 while preserving the number of equations. Equation (Equation 21) can also be rewritten as a linear matrix equation: 
(22)
$$\left[\begin{array}{ccc} \; & \vdots \\ 1 & \frac{x_{T}-x_{i}}{-\lambda ∥p_{T}-p_{i}∥}\; & \frac{y_{T}-y_{i}}{-\lambda ∥p_{T}-p_{i}∥}\\ \; & \vdots \end{array}\right]\left[\begin{array}{c} {\Delta f_{c}}_{T}\\ {v_{T}}_{x}\\ {v_{T}}_{y} \end{array}\right]=\left[\begin{array}{c} \vdots \\ {\Delta \mathrm{CFO}}_{i}\\ \vdots \end{array}\right].$$

which is solvable if there is at least three anchors. The proposed algorithm is summarized in Algorithm 1.
**Algorithm 1:** Velocity estimation **Data**: *N* anchors, 
N≥3
 **Result**: position 
$p_{T}$
 and velocity 
$v_{T}$
 of a moving target1 **foreach** *anchor i* **do**2 
measure the ToA (
$T_{i}\left(t_{B}\right)$
) and CFO (
${CFO_{B}}_{i}$
) of the last broadcaster packet;3 
measure the ToA (
$T_{i}\left(t_{T}+\tau_{i}\right)$
) and CFO (
$CFO_{i}$
) of the current target packet;4 
compute 
${\Delta \mathrm{CFO}}_{i}=CFO_{i}-{CFO_{B}}_{i}$
 (Equation 21)5 
compute 
$t_{i}^{C_{B}S}=\left(T_{i}\left(t_{T}+\tau_{i}\right)-T_{i}\left(t_{B}\right)\right)\left(1+\frac{{CFO_{B}}_{i}}{f_{c}}\right))$
 (Equation 14);6 **end**
(
7 Solve the TDoAs system generated from (Equation 15) using [18] to determine the target location 
$p_{T}$
8 Solve (Equation 22) to estimate the target velocity 
$v_{T}$


### 2.5. Simulation and Experimental Setup

In this section, several setups are described to evaluate the CFO-assisted algorithm. They consist of a TDoA system, as modeled in Section 2.1. Targets and broadcasters send the OFDM legacy preamble (L-STF and L-LTF) of the 802.11 standard [19]. ToAs and CFOs are extracted using the Schmidl and Cox algorithm [20,21]. ToA estimates are further refined by fitting a parabola on the correlation function, as described in [22].

#### 2.5.1. Simulation Setup

The simulations setup is used to perform large-scale simulations with controlled system parameters, as to provide conclusions that can be generalized to any hardware system. This setup aims at simulating the system depicted in Figure 1 and Figure 2 with two receivers. The LO model is the following [23,24]:
(23)
$$\left[\begin{array}{c} \phi (t+T)\\ \omega (t+T) \end{array}\right]=\left[\begin{array}{cc} 1 & T\\ 0 & 1 \end{array}\right]\left[\begin{array}{c} \phi (t)\\ \omega (t) \end{array}\right]+n\left(T\right),$$

where 
ϕ,ω
 are the phase and angular frequency offset of the LO. 
n
 is the process noise vector and follows 
n∼N(0,Q(T))
:
(24)
$$Q\left(T\right)=\omega_{c}^{2}q_{1}^{2}\left[\begin{array}{cc} T & 0\\ 0 & 0 \end{array}\right]+\omega^{2}q_{2}^{2}\left[\begin{array}{cc} T^{3}/3 & T^{2}/2\\ T^{2}/2 & T \end{array}\right],$$

where 
$\omega_{c}$
 is the angular carrier frequency, 
$q_{1}^{2}$
 and 
$q_{2}^{2}$
 are process noise parameters accounting for white and random walk frequency noises. The parameters of the simulation are summarized in Table 1.

#### 2.5.2. Lab Setup

The lab setup aims at providing hardware validation in a controlled environment without suffering from real-world effects, such as multipath, limited SNR, etc. The setup consists of two receivers, one target and one broadcaster, using four USRP-X310 Software Defined Radios (SDRs), as depicted in Figure 4. The SDRs are interfaced using the USRP Hardware Driver (UHD) version 3.9 from Ettus Research, Austin, TX, USA. Signals are sent through cables, and RF splitters are used to broadcast a signal to the two separate receivers. There are 24 measurements, with about 40 s of recording each. The device’s roles as broadcaster, target and anchors are permuted throughout the experiments. The baseband signals are processed to extract ToA and CFO. Then, packets from one receiver are associated with their corresponding packets at the other receiver. The parameter values are summarized in Table 1.

#### 2.5.3. Roadside Setup

Finally, the roadside setup aims to validate the proposed methods and algorithms in a realistic, vehicular environment. The experimental setup is the following: six receivers (USRP-X310 SDR) for TDoA acquisition, one broadcaster (USRP-X310 SDR), and up to four targets (USRP-E310 SDR). Signals are sent Over-The-Air. The RF characteristics are similar to the lab, except the broadcaster transmitting at 2.55 
G

Hz
 carrier (see Table 1). Target Global Positioning System (GPS) positions are used as ground-truth. The experimental environment is shown in Figure 5. During the experiment, no other vehicle than the targets is present.

There are 30 to 40 measurements, 40 s recordings each, for each of the three road scenarios depicted in Figure 6: Major Road (MR), Road Junction (RJ), Roundabout (RA). Target paths are changed every 10 measurements. The processing of the baseband signals is similar to the lab. The extracted features (ToAs with unknown transmission time, CFOs and GPS positions) are further processed according to the availability of the ground-truth, amounting to almost 2 million snapshots. More details regarding the experiment and how the dataset is preprocessed can be found in [18].

## 3. Results

### 3.1. TDoA Estimation

#### 3.1.1. Errors

The proposed synchronization method (Equation 11) is tested on simulation and lab setups. Packets from the broadcaster are dropped at regular intervals to evaluate the estimator across several 
Δt
, i.e., when the time between the target message and the last broadcaster message becomes larger. Each 
Δt
 category is a 20 
m

s
-wide bin. The resulting empiral Cumulative Distribution Function (eCDF) of TDoA errors are shown for simulations and lab experiments in Figure 7a,b, respectively.

The increase in 
Δt
 induces a degradation of the TDoA estimates. Furthermore, the CFO-assisted method performs better than the broadcaster-assisted method in every case. While Figure 7a,b show similar behavior, differences in absolute values are expected for several reasons, such as:Imperfect clock model and different time resolutions, explaining the significant difference at 
Δt=100 ms
.Lack of clock diversity in the lab setup, containing only six LOs in an identical environment. This explains the plateau in broadcaster-assisted results.

#### 3.1.2. Variance

For each lab measurement, the mean and variance of the CFOs at each receiver are estimated from the measured ones. The variance of 
Δt
 is estimated from the time resolution at the receiver, while the mean is estimated by computing the histogram, and then taking the means of each resulting bin.

The upper bound is computed by feeding the parameters described above into (Equation 17). The experimental 
$\sigma_{\tau_{ij}}$
 is estimated by taking the root-mean-square error (RMSE) of every bin. The result is shown in Figure 8b. As expected, the standard deviation linearly increases as the time difference between the target and the broadcaster messages increases. If one can estimate the covariance between CFOs, then a better estimate can be achieved.

Figure 8a shows the effect of CFO covariance in simulation. The more significant the covariance is compared to the measurement noise, the more (Equation 17) overestimates the CFO-assisted TDoA estimator variance.

### 3.2. Localization

The effect of synchronization on location accuracy is evaluated on all roadside measurements. Figure 9 shows eCDFs of the location error for each scenario. For the broadcaster-assisted method, the location accuracy at 
Δt=100 ms
 drops to accuracies similar to the CFO-assisted method at 
Δt=1 s
, while the location accuracy is similar at 
Δt=10 ms
 and 
Δt=100 ms
 for the CFO-assisted method.

### 3.3. Velocity

The performance of the velocity estimator (Equation 22) is evaluated on all roadside measurements. To lower the noise on the CFO estimation, a moving average is applied on the right side of (Equation 22). Their estimated standard deviations are summarized in Table 2. The size of the window is constrained by the constancy of the Doppler frequency (Equation 18) over time. Hence, only window sizes below 500 (5 
s
) are presented. Figure 10 shows the eCDFs of the speed error for each scenario, defined as 
$\left|∥v_{T}∥-∥\hat{v_{T}}∥\right|$
, where 
$v_{T}$
 is the true velocity, 
$\hat{v_{T}}$
 the estimated one. One of the six anchors, in MR and RJ scenarios, has been excluded for the velocity estimation due to analog saturation with the broadcaster.

Lowering 
$\sigma_{\Delta \mathrm{CFO}}$
 increases the accuracy of the speed estimation. However, targets are not moving faster than 55 km/h and, 95% of the time, they do not reach 30 km/h. Hence, getting accurate speed estimations—relative to the target’s low speed—is challenging. eCDFs of the errors of direction, defined as the angle between actual and estimated velocities, are shown in Figure 11. This illustrates that the coarse estimation of the direction of the target is possible. It performs worse in roundabout scenarios. Apart from the higher 
$\sigma_{\Delta \mathrm{CFO}}$
, averaging over several measurements is less effective since it averages measurements with different directions.

## 4. Discussion

The results of Section 3 show that CFO-assisted estimation allows for localization of unsynchronized targets, even when the anchor nodes are not synchronized explicitly. The main advantage of the proposed method is that, unlike existing algorithms, it requires only a single message from the target node. The main drawback of the proposed method is the increased error that occurs when the time difference between the target and broadcaster message becomes large. In the vehicular scenarios investigated in the experiments of this paper, delays higher than 
Δt=100 ms
 make the errors too large in practice.

The theoretical upper bound on the error variance of the proposed algorithm allows system designers to choose the parameters of the localization system, based on the TDoA error that can be tolerated from a design perspective.

The proposed method also allows us to estimate the speed of the target, despite the large CFOs that may occur between target and anchor nodes. In practice, however, the speed estimation errors are too large to be usable in practice with one-message estimation. Reducing the CFO estimation error might lead to lower speed errors, but even lower CFO estimation errors still result in large speed estimation errors (as shown in Figure 10). While the actual speed might not be reliably estimated, the direction of the vehicle can still be estimated with reasonable reliability (as shown in Figure 11), especially if the CFO estimation error can be reduced. This is interesting for vehicular scenarios, where the direction of a vehicle along a road axis might already provide valuable contextual information.

## 5. Conclusions

This paper proposes a CFO-assisted TDoA estimation algorithm, that allows for unsynchronized targets and anchors. The algorithm relies on the fact that CFO and time drift in wireless transceivers have the same physical origin, i.e., the LO offset, and an additional broadcaster.

The proposed method performs well, both in controlled environments and in practical situations, with TDoA errors below a few tens of nanoseconds. The estimator is extended to include speed estimation, but the accuracy is too low to be used for anything more than direction estimation.

Using the CFO to compensate for time drift is a promising method for node synchronization. An extension of this work could be in the context of distributed multi-antenna systems, where time, frequency and phase synchronization are required [24]. Extending the proposed algorithm to distributed systems could also provide interesting developments, especially in Internet-of-Things networks where messages from target nodes are only sent sparsely.

## Figures and Tables

**Figure 1 sensors-24-00737-f001:**
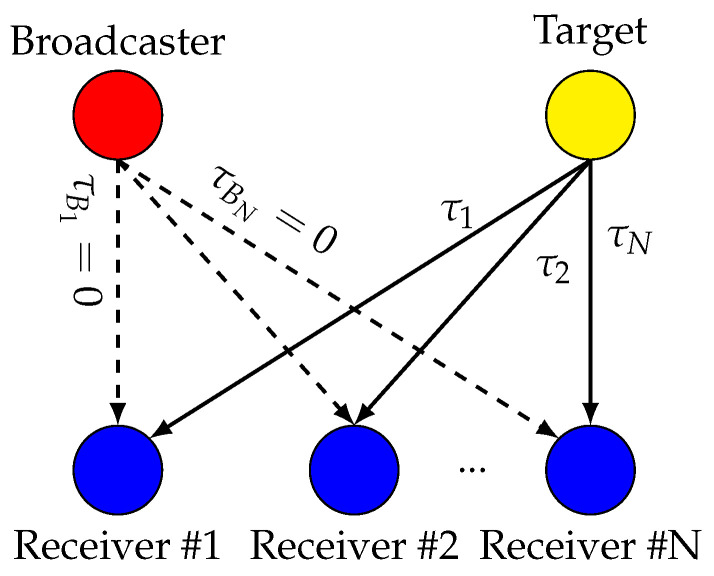
Diagram of a Time Difference of Arrival (TDoA) system. A target sends Radio Frequency (RF) packets that are sensed by N receivers. A broadcaster broadcasts packets to help synchronize the receivers. 
$\tau_{i}$
 represents the propagation delay. Receivers and broadcasters are in known positions, allowing their propagation delays to be compensated.

**Figure 2 sensors-24-00737-f002:**
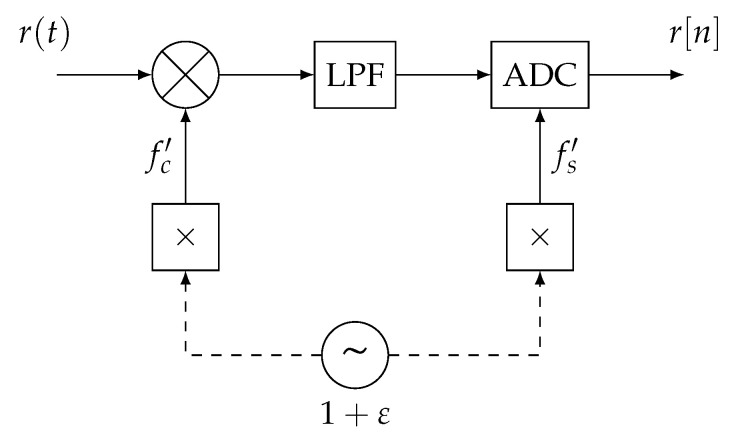
Simplified diagram of an Radio Frequency (RF) receiver’s analog front-end. The ADC and down-converter are driven by the same Local Oscillator (LO).

**Figure 3 sensors-24-00737-f003:**
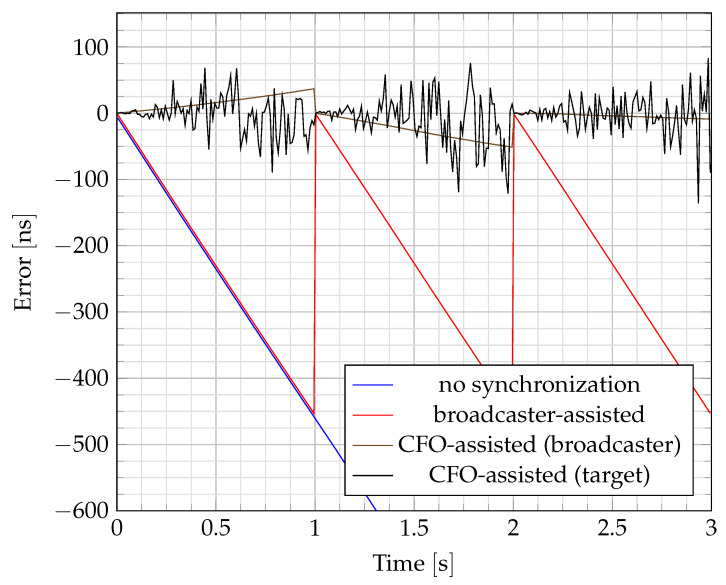
Variation of the Time Difference of Arrival (TDoA) error with different synchronization methods. Without any synchronization, the error increases to infinity (in absolute value). It can be compensated every time a new broadcaster’s message is received (broadcaster-assisted). Between two broadcaster’s messages, the error can be further reduced by using the Carrier Frequency Offset (CFO) of the target or the broadcaster. For illustration purposes, the constant offset 
$\Delta \alpha_{Rx}$
 that should be present without synchronization has been canceled.

**Figure 4 sensors-24-00737-f004:**
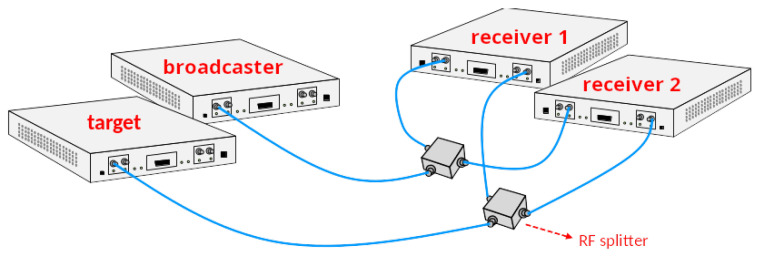
Lab setup. Two USRP-X310 Software Defined Radios (SDRs) send packets through coaxial cables and splitters to two other USRP-X310 SDRs.

**Figure 5 sensors-24-00737-f005:**
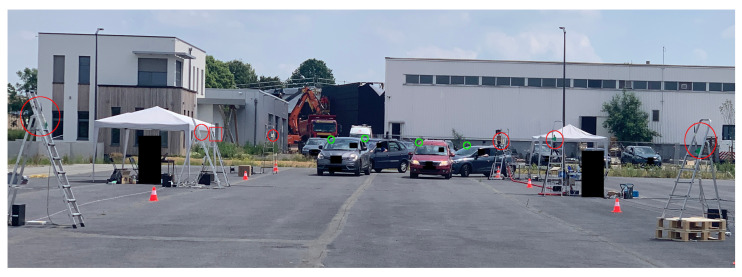
Experimental setup environment. 
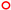
 are the receivers, 
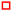
 is the broadcaster and 
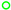
 are the targets.

**Figure 6 sensors-24-00737-f006:**
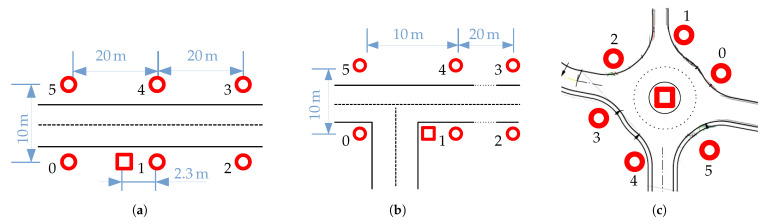
Map for the Over-The-Air (OTA)-synchronized experiment. It combines 6 Time Difference of Arrival anchors (
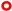
) and 1 broadcaster (
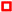
). The devices are placed 
≈2 m
 above ground. (**a**) Map for Major Road (MR) scenarios. (**b**) Map for Road Junction (RJ) scenarios. (**c**) Map for Roundabout (RA) scenarios.

**Figure 7 sensors-24-00737-f007:**
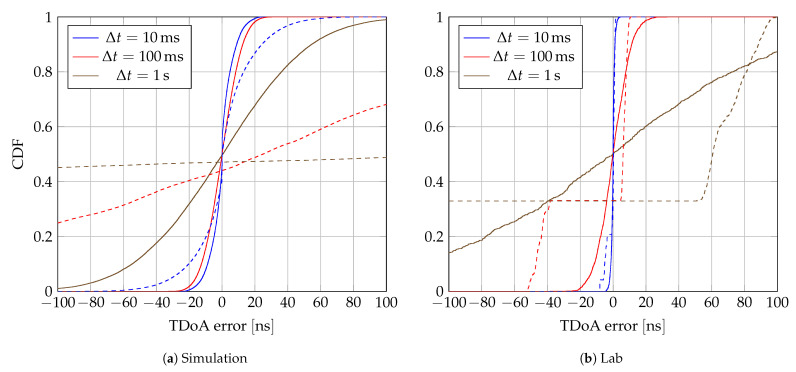
Empiral Cumulative Distribution Function (eCDF) of Time Difference of Arrival (TDoA) errors for several 
Δt
. Dashed lines are for broadcaster-assisted synchronization, solid lines for Carrier Frequency Offset (CFO)-assisted synchronization. 
Δt
 are 20 
m

s
 wide, centered about the mentioned values.

**Figure 8 sensors-24-00737-f008:**
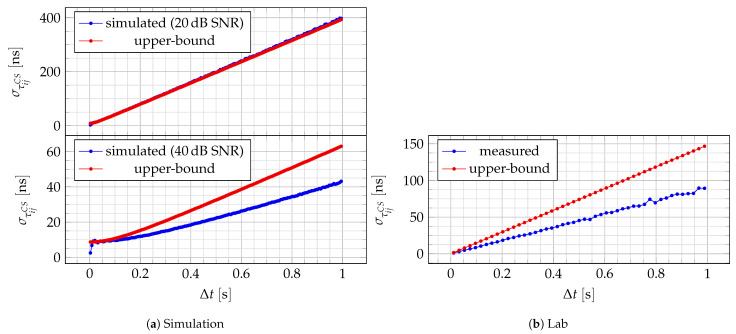
Experimental 
$\sigma_{\tau_{ij}}$
 and its theoretical upper-bound.

**Figure 9 sensors-24-00737-f009:**
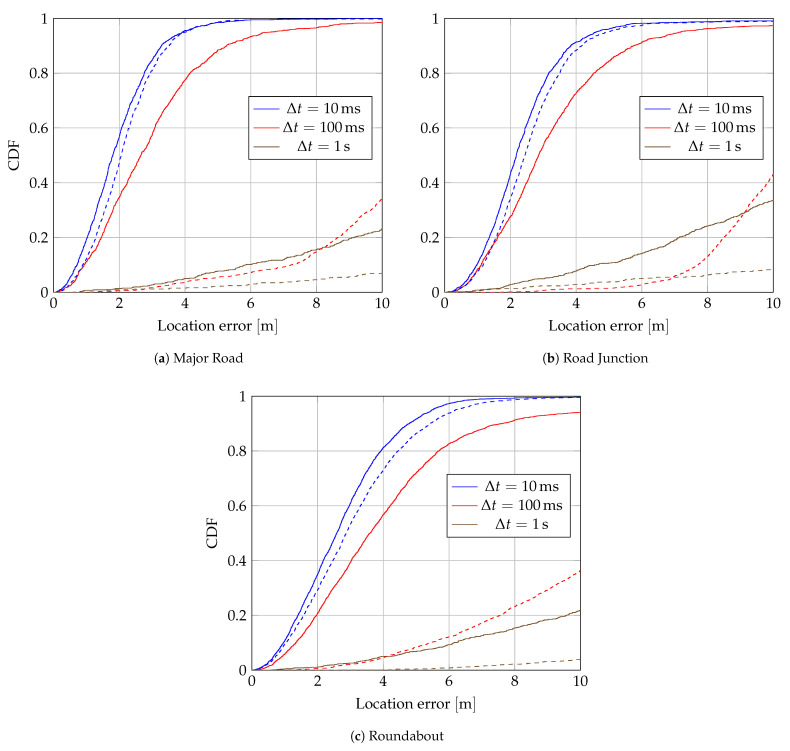
Empiral Cumulative Distribution Function (eCDF) of localization errors for several 
Δt
. Solid lines are for Carrier Frequency Offset (CFO)-assisted synchronization, dashed lines for broadcaster-assisted synchronization. 
Δt
 are 20 
m

s
 wide and centered about the mentioned values.

**Figure 10 sensors-24-00737-f010:**
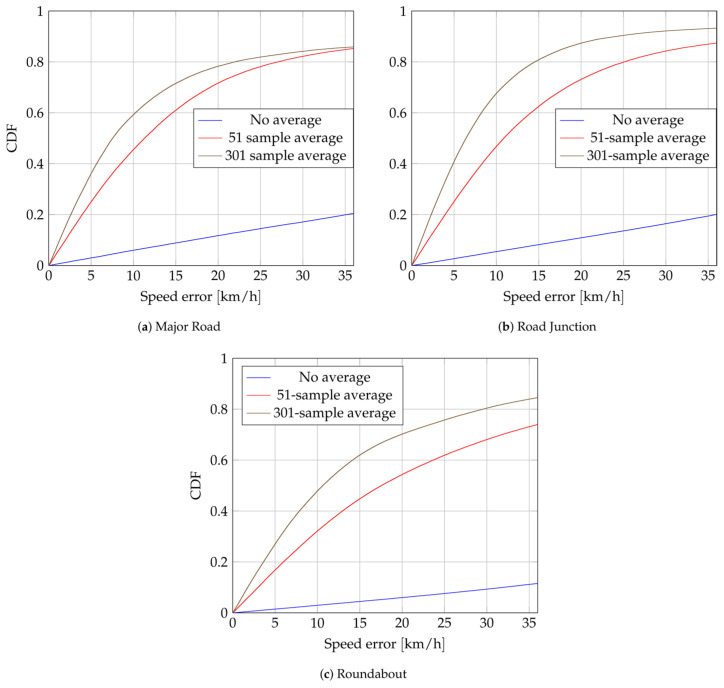
Empiral Cumulative Distribution Function (eCDF) of speed errors for several 
$\sigma_{\Delta \mathrm{CFO}}$
.

**Figure 11 sensors-24-00737-f011:**
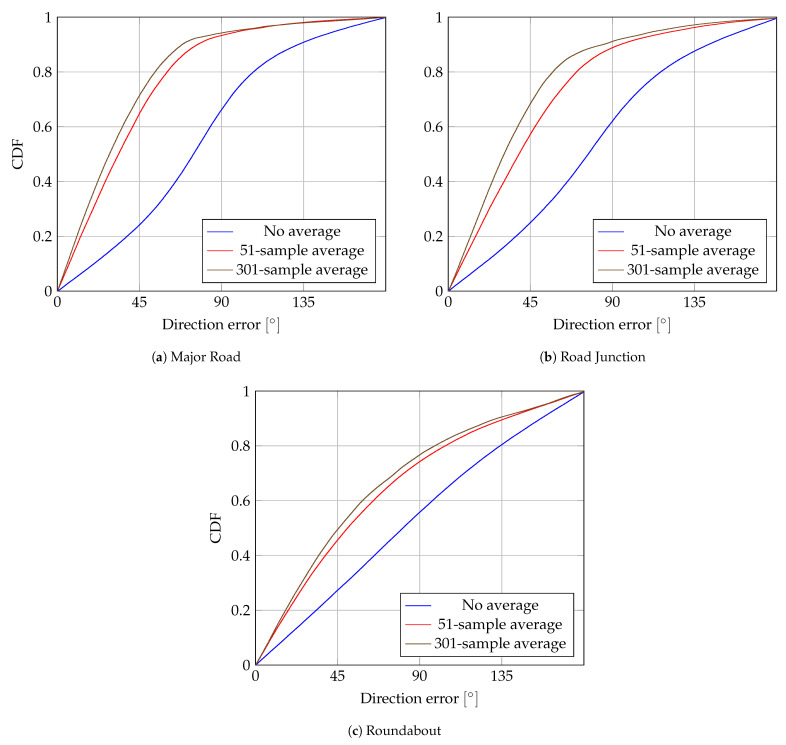
Empiral Cumulative Distribution Function (eCDF) of direction errors for several 
$\sigma_{\Delta \mathrm{CFO}}$
.

**Table 1 sensors-24-00737-t001:** Summary of setup parameters.

	sim	lab	exp
$\epsilon_{Tx,Rx}$	2 μ s / s	2.5 μ s / s	2.5 μ s / s
$q_{1}^{2}$	4.235 × 10^−20^ s	-	-
$q_{2}^{2}$	2.755 × 10^−16^ Hz	-	-
*T*	2 m s	-	-
$T_{\mathrm{target}}$	2 m s	10 m s	10 m s
$T_{\mathrm{broadcaster}}$	1 s	10 m s	10 m s
$T_{\mathrm{run}}$	20 s	40 s	40 s
$N_{\mathrm{run}}$	500	24	100
$f_{c_{\mathrm{target}}}$	2.35 G Hz	2.35 G Hz	2.35 G Hz
$f_{c_{\mathrm{broadcaster}}}$			2.55 G Hz
$f_{s_{Tx}}$	20 M Hz	20 M Hz	20 M Hz
$f_{s_{Rx}}$		100 M Hz	100 M Hz
SNR	40 dB	-	-

**Table 2 sensors-24-00737-t002:** Estimated value of 
$\sigma_{\Delta \mathrm{CFO}}$
.

	MR	RJ	RA
No averaging	560 Hz	440 Hz	610 Hz
51-sample average	90 Hz	80 Hz	110 Hz
301-sample average	50 Hz	40 Hz	60 Hz

## Data Availability

No data are publicly available.

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
