# Peer review of "A CFO-Assisted Algorithm for Wireless Time-Difference-of-Arrival Localization Networks: Analytical Study and Experimental Results"

_sensors, 2024, doi:10.3390/s24030737_

Round 1
Reviewer 1 Report
Comments and Suggestions for Authors
- This manuscript introduces a method for synchronizing receivers' local time in localization systems using time-difference of arrival, by compensating the time difference. It is not mentioned in the manuscript, but the time offsets and drifts are supposed to be invariant for each transmitter-receiver pair.
- Overall, this manuscript is well written. However, the scenarios are rather simple where the time of arrival for each device pair is linear invariant (offset and drift parameters are constant). In reality, it is much dependant on the transmission environment which is varying, and also on the obstactles that may appear.
- The time imperfection model given in Eq. (3) is mentioned as common, but still there would be a reference.
- For the Roadside experiment, the duration of experiment and some other conditions are not mentioned, such as whether there are vehicles passing during the data collection period.
Reviewer 2 Report
Comments and Suggestions for Authors
The paper presents a TDoA estimation algorithm in which unsynchronized targets and anchors can compensate the time and CFO offsets by means of a single message from a broadcaster node. The method and the models on which it is based are described clearly, and the results are supported by simulations and experimental results.
The main issue is the lack of a comparison, in the numerical results, with a standard TDoA system, in which wireless synchronization among receivers is achieved with other methods, for instance a GNSS reference or a standard broadcaster.
